# The Role of the CD28 Family Receptors in T-Cell Immunomodulation

**DOI:** 10.3390/ijms25021274

**Published:** 2024-01-20

**Authors:** Klaudia Ciesielska-Figlon, Katarzyna A. Lisowska

**Affiliations:** Department of Physiopathology, Medical University of Gdansk, 80-211 Gdansk, Poland; klaudia.ciesielska-figlon@gumed.edu.pl

**Keywords:** T cells, CD28 family receptor, B7 family of ligands, ICOS, CTLA-4, PD-1, BTLA

## Abstract

The CD28 family receptors include the CD28, ICOS (inducible co-stimulator), CTLA-4 (cytotoxic T-lymphocyte antigen-4), PD-1 (programmed cell death protein 1), and BTLA (B- and T-lymphocyte attenuator) molecules. They characterize a group of molecules similar to immunoglobulins that control the immune response through modulating T-cell activity. Among the family members, CD28 and ICOS act as enhancers of T-cell activity, while three others—BTLA, CTLA-4, and PD-1—function as suppressors. The receptors of the CD28 family interact with the B7 family of ligands. The cooperation between these molecules is essential for controlling the course of the adaptive response, but it also significantly impacts the development of immune-related diseases. This review introduces the reader to the molecular basis of the functioning of CD28 family receptors and their impact on T-cell activity.

## 1. Introduction

The overarching function of the immune system is to protect the body against pathogenic microorganisms and cancer. The proper response of this system depends on effector elements such as cells (lymphocytes or phagocytic cells) and proteins, such as antibodies, complement components, and acute-phase proteins. The immune system can recognize and eliminate harmful agents such as bacteria, viruses, parasites, toxins, and even abnormal cells showing signs of tumorigenesis. The immune system identifies foreign molecules (antigens) and manifests immune memory, which speeds up the process of removing dangerous agents when they come into contact again. However, for the immune system to perform all these functions and function properly, a sequence of activation of its various components must occur.

In the broad concept of immunomodulation, the activation of T cells and the specific role of the CD28 molecule are of great importance. The CD28 family receptors include the CD28, ICOS (inducible co-stimulator), CTLA-4 (cytotoxic T-lymphocyte antigen-4), PD-1 (programmed cell death protein 1), and BTLA (B- and T-lymphocyte attenuator) molecules. They characterize a group of receptors and ligands similar to immunoglobulins that control the immune response. Among the family members, CD28 and ICOS function as enhancers of T-cell activity, while three others—BTLA, CTLA-4, and PD-1—function as suppressors. Proteins from the B7 family of ligands engage the receptors of the CD28 family. These ligand–receptor pairs that exert inhibitory or stimulatory effects on immune responses are immune checkpoint molecules, crucial for maintaining self-tolerance and modulating the length and magnitude of immune responses.

This review presents the current state of knowledge on the physiological role of receptors of the CD28 family in T-cell activation and their immunomodulatory capabilities.

## 2. T-Cell Activation

T-cell activation is one of the essential processes in the human immune system by which a naive T cell, recognizing a specific antigen, can differentiate into various effector cells, aiming to eliminate the foreign moiety antigen. The proper lymphocyte activation thus underlies a normal response against pathogens, and its defects can cause immune disorders, for example, hypersensitivity reactions or immunodeficiencies [1].

Mature lymphocytes that leave the thymus have the phenotype of naive T cells [2]. To fully participate in the immune response, they require activation. The condition for activating a naive T cell is that there should be direct contact through the T cell antigen receptor (TCR) with a specific major histocompatibility complex (MHC) molecule of an antigen-presenting cell (APC). Most often, this process takes place in the peripheral lymphoid organs. If antigen recognition occurs, the lymphocyte adheres more firmly to the APC, and then the next activation steps can initiate the formation of an immune synapse [3].

The presence of the CD8 or CD4 molecule determines the type of tissue compatibility system molecules with which T cells interact. The presentation of antigens in combination with MHC class I molecules leads to antigen recognition by cytotoxic T cells (Tc cells or CTLs), which have the CD8 protein on their surface. Antigens combined with MHC class II molecules are generally presented to helper T cells (Th cells), which have the CD4 protein on their cell membrane [4]. The activation of lymphocytes is a several-step process: at first, there is specific antigen binding by the TCR receptor, followed by stimulation of non-receptor tyrosine kinases, activating effector proteins, and the MAPK (mitogen-activated protein kinase) cascade. As a result of the cascade of excitation of individual factors that are part of the lymphocyte activation pathway, transcription factors required for full lymphocyte activation are activated [5].

### The Molecular Aspects of T-Cell Activation

For full lymphocyte activation, the cell must receive at least two, and optimally three, signals. The first activation signal comes from antigen recognition by the TCR. The second activation signal comes from co-stimulatory molecules; in the case of T cells, it is the interaction between CD28 family receptors and the B7 family of ligands (Figure 1) [1]. The activation process can be modified by a third signal coming from receptors for cytokines [6]. Unlike naive cells, T cells are already activated, and memory cells require only the first signal to start proliferation [7].

A stable association of T cells with the APCs is achieved through additional protein interactions. An important role in this process is the CD2/LFA-1 (lymphocyte function-associated antigen 1) interaction [8] and the membrane integrins LFA-1 and VLA-4 (very late antigen-4), which interact with adhesion proteins such as VCAM-1 (vascular cell adhesion molecule 1) and ICAM-1 (intercellular adhesion molecule 1) [9]. Identifying the presented antigen (the first signal) is insufficient for full T-cell activation. In the absence of the second signal (from a co-stimulatory molecule), the cell is launched into anergy [10].

The binding of the antigen and MHC molecules by the extracellular part of the α and β chains of the TCR complex initiates the transmission of the signal to the interior of the cell [4]. In this process, protein phosphorylation plays a significant role, but the TCR complex does not have enzymatic properties, so it is necessary to involve non-receptor kinases such as LCK (lymphocyte-specific protein tyrosine kinase) [11]. In the early stages of signal transduction, phosphorylation of tyrosine residues occurs within the chains of the CD3 complex. Phosphorylated tyrosine residues form docking sites for ZAP-70 (zeta-chain-associated protein kinase 70), which is necessary for the normal processes of lymphocyte maturation in the thymus and activation of peripheral T cells. The result of phosphorylation is a cascade of activation of proteins responsible for signal transduction into the cell [12].

ZAP-70 can phosphorylate tyrosine residues found in the LAT (linker for activation of T cells) protein. It is an adaptor protein bound to the cell membrane with no enzymatic properties [12]. Instead, it is a molecular link between surface receptors and molecules participating in subsequent signal transduction steps. Phosphorylated tyrosine residues within the LAT are a type of molecular platform that enables the recruitment of proteins relevant to the T-cell activation pathway [13]. Recruitment of PLCγ-1 (phospholipase Cγ-1) is obligatory to activate the pathway associated with phosphatidylinositol metabolism. On the other hand, the recruitment of GADS (Grb2-related adaptor downstream of Shc) and SLP-76 (lymphocyte cytosolic protein 2) proteins, which are constitutively bound, is essential for the important process of rearrangement of the cytoskeleton of the activating cell [14].

Recruitment of the GRB-2 (growth factor receptor-bound protein 2) protein found in the complex with the SOS (Son of Sevenless) protein is necessary for the MAPK cascade. The MAPK cascade is initiated by proteins with the ability to bind and dephosphorylate guanosine triphosphate (GTP) [15]. The activated form of these proteins is the GTP-bound form, so agents that increase their GTP-ase activity are inhibitors of the transmitted signal, and agents that promote the exchange of guanosine diphosphate (GDP) for GTP are stimulators [15]. The RAS (rat sarcoma) protein is a GTPase bound to the cell membrane on the cytoplasmic side of the T cell, and the protein transmits the signal through a cascade of serine/threonine kinases to the cell nucleus. The RAS, in complex with GTP, is active and recruits the RAF (rapidly accelerated fibrosarcoma) kinase [16]. Subsequent phosphorylation of proteins involved in the lymphocyte activation pathway leads to the activation of the transcription factor ELK-1 (ETS-like-1 protein). The activated ELK-1 factor is translocated to the cell nucleus, where it activates the expression of the FOS protein [17]. The FOS protein forms a heterodimer with JUN, forming the AP1 (activator protein 1) complex, which has a binding site in the promoter region of the gene encoding IL-2 [18].

After antigen recognition by the TCR, PLCγ-1 rapidly translocates to the vicinity of the TCR complex, where it is activated by tyrosine kinase-mediated phosphorylation [5]. The activation results in the catabolism of cell membrane phospholipids, mainly phosphatidylinositol diphosphate (PIP2) [19]. As a result, second-order transmitters are formed. Inositol 1,4,5—triphosphate (IP3) and diacylglycerol (DAG) are the most important in this process. The role of IP3 is to release calcium ions from intracellular stores. Then, with the involvement of calmodulin, a serine phosphatase called calcineurin is activated. The target molecules for calcineurin are transcription factors are categorized as nuclear factors of activated T cells [20]. These factors are abbreviated as NF-AT (nuclear factor of activated T cells), and in resting T cells, they are located in the cytoplasm [21]. After activation by calcineurin, they move to the cell nucleus, where they form a complex that plays a vital role in the transcription of genes undergoing expression in the activated T cells.

The process of T-cell activation is very dynamic. At the same time, DAG directly activates serine/threonine kinases of the protein kinase C (PKC) family. These kinases play an important role in activating factors belonging to the family of nuclear factor kappa-light-chain-enhancer of activated B cells (NF-κB) factors [21]. Activated by phosphorylation, NF-κB factors are then translocated into the cell nucleus, where they play a crucial role in regulating the expression of genes related to the lymphocyte activation process. A non-negligible role in this process is played by dynamic changes in the cytoskeleton rearrangements regulated by the VAV-1 (vav guanine nucleotide exchange factor 1), NCK (non-catalytic region of tyrosine kinase adaptor protein 1), and RHO (rhodopsin) proteins [22].

## 3. Interactions between CD28 Family Receptors and B7 Family of Ligands

Complete activation of a T cell occurs only after it receives two signals, so in this publication, we emphasize the importance of co-stimulatory signaling. Co-stimulatory signals regulate various T-cell functions, both positively and negatively [10]. These functions are controlled by specific signaling pathways and the expression of receptors and their ligands, which are dynamically and quantitatively regulated on T cells and APCs, respectively. The immune synapse is the control site for the transmitted signal from the TCR complex [23]. The initiators of this control are two groups of molecules: co-stimulatory and inhibitory. The balance between the influence of both types of molecules is crucial for the proper functioning of the immune system. Based on this information, the concept of immune system checkpoints has recently been developed [23].

The involvement of co-stimulatory molecules is essential for achieving optimal T-cell activation. An effective T-cell response hinges on maintaining equilibrium between stimulating and inhibitory signals. The process occurs thanks to interactions between CD28 family receptors and the B7 family of ligands. Over the past two decades, several molecules within the CD28 and B7 families have been identified [24]. The CD28 receptor family includes CD28, CTLA-4 (CD152), PD-1 (CD279), ICOS (CD278), and BTLA, which are expressed on T cells. CD80 (B7-1) and CD86 (B7-2) are the two main ligands belonging to the B7 family of ligands present in the APCs [25]. These molecules can transmit positive or negative signals, in addition to the antigen-specific signal, to T cells [26].

### 3.1. CD28 Receptor as an Example of a Co-Stimulatory Molecule

The CD28 molecule was described by Hansen in 1980 as a T-cell surface antigen. A study from 2005 proved that the extracellular fragment of CD28 shows a structure characteristic of the immunoglobulin superfamily [27]. CD28 is a 44 kDa glycoprotein with a homodimer structure. The structure is composed of a V-shaped extracellular domain with disulfide bonds and a MYPPPY motif, which is required for B7 ligand binding, as well as a membrane part and a cytoplasmic domain composed of 41 amino acids [28]. The gene encoding the molecule is 300 kb in size and is located on chromosome 2 in the region of the q33 strand [29]. The molecule is constitutively present in 30–50% of human CD8+ T cells and 95–100% of CD4+ T cells, with 60,000 molecules per cell [30]. CD28 expression increases after binding to the TCR/CD3 complex. When B7 is bound to CD28 during T-cell activation, the mRNA level of the CD28 molecule is decreased, as is surface expression. This inhibitory regulation is transient and lasts up to 48 h. It prevents the restimulation of T cells, which affects the extent and duration of the immune response [31].

CD28 plays a role in various T-cell activities, such as reorganizing the cell structure, generating signaling molecules like cytokines and chemokines, and facilitating internal biochemical processes like phosphorylation, transcriptional signaling, and metabolism. These processes are crucial for T-cell growth and specialization. Activation of CD28 receptors induces changes in T cells at epigenetic, transcriptional, and post-translational levels. One notable effect of CD28 co-stimulation is its control over several functions in T cells, including activating cytokine genes. One specific gene encodes IL-2, a cytokine that impacts T-cell survival, growth, and differentiation. When CD28 co-stimulation is absent, IL-2 production decreases, making T cells unresponsive to stimulation [32]. Additionally, CD28 activation leads to a type of protein modification called arginine methylation in multiple proteins [24].

The formation of an immune synapse initiates the activation of lymphocytes. Within it, there is a concentration of TCR receptors, and their connection to MHC complexes is made possible by the interaction of adhesion molecules. It is now known that the signal from CD28 enhances the initial fusion of T cells and APCs. The CD28 molecule participates, thus, in cellular adhesion, proving its high significance for the early stages of activation. By regulating IL-2 expression, CD28 indirectly affects T cell proliferation [33].

Signaling from the TCR complex does not always stimulate cell activation but can redirect the cell into a state of anergy or even programmed cell death. For apoptosis to occur, a large amount of antigen on the T cell and the absence of pro-inflammatory factors, including anti-apoptotic cytokines (IL-2, IL-7, and IL-15), are required [34]. Apoptosis induced by the interactions between Fas (CD95) and its ligand FasL (CD95L) is crucial for the elimination of T cells in the final phase of the immune response. This type of apoptosis is called activation-induced cell death (AICD). The CD28 molecule protects the cell from Fas-induced apoptosis as the signaling pathway involving the kinases Akt and PI3K (phosphoinositide 3-kinase) is activated. Activated Akt (also known as protein kinase B) inhibits the progression of apoptosis by suppressing the recruitment of caspase 8, which is crucial in cell death [35]. CD28 also promotes lymphocyte survival by upregulating Bcl-xL expression and reducing cell membrane permeability [36].

Lymphocyte maturation is a several-step process, with only 10% of properly selected cells reaching the periphery. Generally, the maturation process can be distinguished by an early phase, positive and negative selection [37]. It has been proven that the CD28 molecule is important during positive selection when double-positive lymphocytes have CD4 and CD8 antigens. This selection verifies whether rearrangements of genes encoding α and β subunits have resulted in the emergence of TCRs capable of recognizing cellular antigens. In the process of positive selection, the CD28 molecule plays a significant role. It has been shown in an experimental model that the absence of CD28/B7 costimulation leads to increased selection in the thymus. This is because the CD28 molecule inhibits the differentiation of mature single-positive T cells by suppressing their selection during differentiation in the thymus [38]. In contrast, negative selection (clonal deletion) leads to the removal of cells that recognize their own MHC class antigens with increased affinity, which could result in autoreactivity [38]. Two distinct pathways are likely involved in negative selection in vitro. The first is CD28-dependent for interactions with low binding capacity, and the second is CD28-independent for interactions with high affinity [39]. Thymocytes primarily need signals that inhibit their apoptosis to survive. Signaling from the CD28 molecule enhances Bcl-XL expression, which in turn prevents Fas-induced cell death [38]. Despite this thorough double-selection process, not all autoreactive lymphocytes are eliminated, as not all existing autoantigens will be presented to lymphocytes in the thymus. Therefore, mechanisms that determine peripheral tolerance are necessary. For example, T-cell activity can be inhibited by increased expression of the CTLA-4 antigen, which transmits an inhibitory signal [40].

### 3.2. ICOS (CD278), Another Co-Stimulatory Molecule

In humans, the ICOS gene is located on chromosome 2q33.2, contains 199 amino acids, and encodes for a protein known as an immune checkpoint protein. It was the third CD28 family member to be identified42. Unlike other family members, ICOS is expressed in already activated T cells and peripheral tissues [41].

In contrast to CD28, the expression of ICOS displays greater variability. ICOS expression is dependent on the activation of T cells through the TCR/CD3 complex. Following activation, ICOS expression remains present on recently activated CD4+ T cells, as well as on memory Th1 and Th2 cells. Besides the TCR signal, the cytokines IL-12 and IL-23 boost ICOS expression in T cells [42]. While it is not obligatory for CD28 to be co-engaged to induce ICOS expression, CD28 can enhance ICOS levels [43]. Although ICOS shows structural similarity to CD28 and CD152, it lacks the specific MYPPPY motif present in CD28 and CTLA-4, which is essential for interaction with CD80 and CD86 [44,45]. Therefore, a ligand for this molecule is ICOSL (B7H, B7RP-1, and CD275) expressed on APCs [46].

The pathway involving ICOS and ICOSL offers a crucial co-stimulatory indication, supporting the expansion of T cells and primarily ensuring their survival. Furthermore, ICOS manages the formation and reaction of T follicular helper (Tfh), Th1, Th2, and Th17 cells and contributes to sustaining the equilibrium of memory effector T cells and regulatory T cells (Tregs) [47]. Indeed, ICOS has a more pronounced capability to initiate the PI3K/Akt pathway and activate the subsequent MAPK cascade when compared to CD28 [48].

### 3.3. CTLA-4 (CD152), a Negative Regulator of T-Cell Activation

The CTLA4 gene is located within strand 2q33 on chromosome 2 in humans. It consists of 233 amino acids [49]. CTLA-4 appears on the surface of activated T cells upon contact with antigen and inhibits further lymphocyte responses. Thus, CTLA-4 provides a negative feedback signal in the specific immune response, preventing it from expanding [50]. Ligands for CTLA-4 found on the surface of APC are CD80 and CD86 from the B7 family of ligands [51]. An unusual feature of the CTLA-4 dimer, rarely seen in other polymeric proteins, is that the ligand-binding sites are located not at the junction of two subunits, jointly forming a single binding site, but at a considerable distance from each other. This allows two ligands to be bound by a single CTLA-4 dimer and consequently allows polymeric CTLA-4 structures to be formed in the cell membrane, alternating with CD80/CD86 in a zip-like pattern [52]. The inhibitory role of CTLA-4, especially for CD28-induced signaling, is evident in mice possessing a defective, non-functional CTLA-4 gene; these animals die at 2–3 weeks of age due to uncontrolled lymphocyte division, leading to massive inflammation in most organs, which is known as lymphoproliferative syndrome [53,54].

The mechanisms of CTLA-4’s inhibitory action are based on specific molecular pathways. Both CD28 and CTLA-4 can bind the same ligands, but CTLA-4 has a higher affinity for CD80 and CD86, which results from the specific binding of dimers described above [55]. As a result, CD28 is displaced from the ligand complexes, and the lymphocyte-activating signal is attenuated [56].

Blocking of the TCR/CD3 complex-derived signal occurs due to the interaction of the intracellular CTLA-4 fragment with the amino acid motif YVKM with SHP-2 phosphatase (tyrosine phosphatase 2) indirectly through PI3K [57]. SHP-2 has the inhibitory properties of phospholipase C gamma 1 (PLCγ1) activated by CD3 [58]. In addition, TCR-mediated signaling can be inhibited by the phosphatase PP2A (protein phosphatase 2A), which likewise binds to CTLA-4 [59].

Inhibition of CD28 co-stimulatory signaling also occurs through PP2A, which can also inhibit CD28-enhanced IL-2 production by blocking Akt kinase [40]. It affects TCR-activated MAPK pathways by stimulating the JNK (c-Jun N-terminal kinase) pathway while inhibiting the ERK (extracellular signal-regulated kinase) pathway [60]. CTLA-4-derived signaling alters the expression of cell cycle control proteins. This is due to an early exit from the G1 phase, entry into the S phase, a prolonged S phase period, and inhibition induced by increased p27 (kip1) expression. The lack of CTLA4 translates into an increase in T-cell proliferation through an increase in IL-2 secretion in the S and G2-M phases [61].

### 3.4. PD-1 (CD279)

The PD-1 receptor is a protein encoded in humans by the PDCD1 gene, whose locus is 2q37.3. It consists of 288 amino acids [62]. It is a receptor that is expressed on T cells, B cells, monocytes/macrophages, dendritic cells (DCs), and, although at a minimal level, in natural killer T (NKT) cells [63]. When it binds to PD-L1 (B7H-1) and PD-L2 (B7H-2), as well as other members of the B7 family of ligands, it inhibits immune system stimulation [64].

Negative co-stimulatory signals conveyed by PD-1 and CTLA-4 serve distinct roles. While CTLA-4 modulates the priming of T cells in lymphoid organs, PD-1 primarily regulates inflammatory responses in peripheral tissues. Furthermore, unlike CTLA-4, PD-1 can hinder TCR- and CD28-triggered activation by enlisting inhibitory phosphatases like SHP-2, which dampens the initiation of PI3K activity [63]. Inhibition of T-cell activity occurs through the dephosphorylation of PI3K, which leads to the blocking of Akt activity and consequently impairs the energy metabolism of the cell [40].

Another effect induced by PD-1 is to block the binding of ZAP-70 kinase to the CD3ζ subunit by preventing the phosphorylation of both molecules. PD-1 consequently blocks TCR receptor-derived signal transduction. PD-1 also inhibits the activation of PKCθ and ERK kinase [65]. It has been proven that the inhibitory effect of PD-1 is reversible when CD28 stimulation occurs and STAT5-activating cytokines, for example, IL-2, IL-7, and IL-15, are released [66,67].

Interactions between PD-1 and its ligands transmit an inhibitory signal to T cells, which has the measurable effect of hampering the proliferation and production of cytokines by these cells [68]. However, studies show that stimulating PD-L1 can induce the conversion of naive CD4+ T cells to induced T regulatory cells (iTregs), which are critical mediators of peripheral tolerance that actively suppress the formation of effector T (Teff) cells [69]. In the absence of PD-1 signaling, PD-L1 and PD-L2 may provide a positive signal to T cells and stimulate their proliferation and cytokine production [70]. Teff cells, when activated, express both PD-1 and PD-L1. Also, higher PD-L1 expression is correlated with a higher Teff proliferation capacity [71].

### 3.5. BTLA

The B- and T-lymphocyte attenuator (BTLA) is another CD28 family member functioning as a negative co-stimulatory receptor. It has been observed that BTLA is consistently expressed at low levels in various cell types, including naïve B and T cells, Tfh cells, macrophages, DCs, NKT cells, and natural killer (NK) cells. However, opinions are divided on the presence of the receptor on Tregs. While some argue that BTLA is found in Tregs [72,73], others deny it [74,75,76].

The BTLA gene is situated in the q13.2 region of chromosome 3 and comprises 5 exons, totaling 870 base pairs in length. The structure of BTLA bears resemblance to that of PD-1 and CTLA-4. BTLA is a type I transmembrane glycoprotein and a member of the immunoglobulin superfamily (IgSF), consisting of 289 amino acids [77]. This structure encompasses an extracellular domain, a transmembrane domain, and a cytoplasmic domain [78].

BTLA binds with the herpes virus entry mediator (HVEM), which does not belong to the classic B7 family. HVEM is a member of the tumor necrosis factor receptor (TNFR) superfamily that has been identified as a BTLA ligand [79]. Its binding with BTLA reduces cell activation, cytokine production, and proliferation [80]. Therefore, BTLA deficiencies in various experimental animal models promote the development of autoimmune diseases or worsen their course [81,82]. For example, BTLA-deficient mice presented more pronounced experimental autoimmune encephalomyelitis.

It should also be emphasized that HVEM can interact with BTLA in a cis or trans manner. HVEM and BTLA can be co-expressed on the same cells, forming a cis complex, or present on different cells, forming a trans interaction [83]. Both types of interaction inhibit T-cell activity, but in different mechanisms; the cis complex prevents interaction with other co-signaling molecules, while trans interaction inhibits nuclear factor kappa-light-chain-enhancer of activated B cells (NF-κB)-dependent cell activation.

The role of BTLA as a negative co-stimulatory receptor has been substantiated by studying mice lacking BTLA, which displayed increased susceptibility to autoimmune disorders [84]. Additionally, laboratory observations have demonstrated that anti-BTLA agonists convey inhibitory signals to T cells [85]. When BTLA is engaged, it hinders the activation of T cells mediated through CD3/CD28. BTLA transmits signals by recruiting the SHP-1 and SHP-2 phosphatases. The activation of SHP-1 and SHP-2 initiates a suppressive signal directed at TCR transduction, effectively halting T-cell activation. A notable distinction in signaling between BTLA and PD-1 is that BTLA brings in SHP-1, which is a more potent phosphatase for inhibiting TCR and CD28 signaling. In the literature, one can find information about bidirectional BTLA signaling. The GRB-2 motif recruits the PI3K protein subunit p85 and further stimulates the PI3K/Akt signaling pathway, which promotes B- and T-cell proliferation [86].

BTLA at the molecular level interacts with proteins involved in the production of IFN-γ (interferon), IL-17RA (IL-17 receptor antagonist), or the development of B and T cells [87], as well as with bone morphogenetic protein (BMP), an intercellular signaling molecule responsible for growth and differentiation pathways. Studies have shown that BMP initiates the p38 MAPK pathway to produce INFs type I (INF-α and INF-β) [88]. If the DDX19B pathway relating to the response to viruses by the RIG-I-like receptors (RLR) is induced, there will be a negative regulation of INF production [89].

The list of CD28 receptor family members and their molecular interactions are presented in Table 1.

## 4. Immunotherapy Based on the Immunomodulation Process

CD28 receptor family members are believed to be attractive targets for immunotherapy, in which they can inhibit or enhance T-cell responses. In autoimmune diseases, silencing immune system activation could be achieved by stimulating silencers, for example, CTLA-4. In anti-cancer therapies, blocking silencers for precise activation of cell pathways specialized to fight cancer would be essential. Currently, drugs based on immune checkpoints are mainly used as anti-cancer drugs.

In recent years, immunotherapies have rapidly developed using the blockade of receptors or ligands of immune checkpoints. The first immune checkpoint inhibitor (ICI) was ipilimumab, a human monoclonal antibody (mAb) that blocks CTLA-4 and enhances the anti-tumor T-cell response. In clinical trials, the target group was melanoma patients who achieved disease stabilization [90]. Standard treatment for some cancers also includes monoclonal antibodies specific for PD-1 or PD-L1. Among the antibodies targeting PD-1 approved by the Food and Drug Administration (FDA) are pembrolizumab (Keytruda), nivolumab (Opdivo), and cemiplimab (Libtayo) [91]. The latest antibody registered in 2023 is toripalimab (Loqtorzi), a monoclonal-specific antibody directed to block PD-1. Its use is possible in the first-line treatment of adults with metastatic or recurrent locally advanced nasopharyngeal cavity cancer [92]. Among the antibodies targeting PD-L1, we have avelumab (Bavencio) and durvalumab (Imfinzi) registered for patients with urothelial carcinoma [91]. The list of ICIs and the diseases for which they are approved is presented in Table 2.

For autoimmune diseases, there is only one checkpoint stimulator, abatacept, a CTLA4-Ig fusion protein, one of the therapeutic options for rheumatoid arthritis. Abatacept attenuates T-cell activation because it regulates T-cell activation by inhibiting the CD80/86-CD28 co-stimulatory pathway, which is required for normal T-cell activation (Table 2) [93].

The importance of CD28 co-stimulatory signals in T-cell function makes this molecule an exciting target for drug design to modulate the function of both effector T cells and Treg cells. Unfortunately, the first attempts to create a soluble CD28 (sCD28) to block co-stimulatory signals failed due to the too-low affinity of sCD28 for ligands [94]. Considering the known mechanisms, attempts are still being made to test available drugs for the treatment of autoimmune diseases [24,95,96], including type 1 diabetes [97], lupus nephritis [98], asthma [99], or Crohn’s disease [100].

Understanding the molecular mechanisms responsible for regulating immune checkpoints may constitute the basis for new immunotherapies. Using small molecules, such as peptides, as adjuvants is another exciting option, as these compounds have high selectivity and potency, better tolerability, and predictable metabolism. An example is glycoprotein D, which can bind with HVEM and block its interactions with BTLA [101]. Another advantage of using ICIs is the possibility of using them in patients who do not qualify for clinical trials for various reasons, for example, cancer patients with pre-existing autoimmune diseases, who are usually excluded from immunotherapy clinical trials due to the high risk of serious adverse events. There is growing evidence that immune checkpoint inhibitors may be safe and effective in this patient population [102].

## 5. Conclusions and Future Perspective

The central importance of CD28 family receptors in T-cell function makes them a tempting target for drugs that modulate the function of both effector T cells and Tregs. These signals are crucial in many T-cell processes, including cytoskeletal remodeling, cytokine production, survival, and differentiation. CD28 and ICOS ligation leads to unique epigenetic, transcriptional, and post-translational changes in T cells that cannot be reproduced by TCR ligation alone, making them crucial for T-cell survival, activation, and maintenance of immune homeostasis. However, the second aspect of controlling T-cell activation is inhibiting the processes that may contribute to immune-mediated tissue damage.

By modulating the length and magnitude of immune responses, immune checkpoint molecules maintain immune homeostasis. Disrupting their function impairs the control of T-cell responses, which may promote the development of autoimmune diseases or cancer. Genetic polymorphisms in human immune checkpoint molecules account for additional variability in this system. Therefore, targeting the receptors or ligands of immune checkpoints using agonistic or antagonistic antibodies can alter immune tolerance and explain the likelihood of adverse reactions and their mechanisms. Expanding knowledge of the function of immune checkpoint molecules and the impact of their polymorphisms can provide information on how to precisely and deliberately modulate T-cell activation pathways to either suppress pathological responses promoting autoimmunity or activate the anti-tumor response.

To fully exploit the possibility of interfering with and controlling the process of T-cell activation, it is necessary to have a better understanding of signaling pathways, relationships between molecules, and the molecular basis of the T-cell activation process. Therefore, it seems reasonable to investigate the signaling pathways of the CD28 family receptors further, which would help propose new biological therapies for autoimmune and cancer diseases.

## Figures and Tables

**Figure 1 ijms-25-01274-f001:**
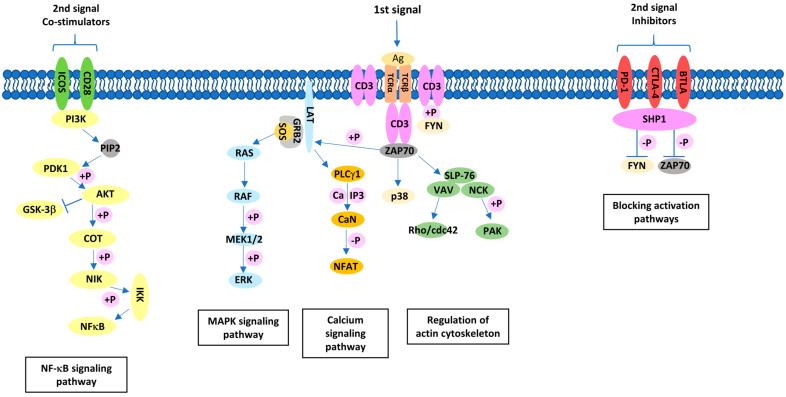
T-cell activation pathway involving co-stimulatory molecules.

**Table 1 ijms-25-01274-t001:** Characterization of CD28 receptor family members.

CD28 Family Receptor	Cellular Expression	B7 Family of Ligand	Molecular Interactions	Type of Regulation
CD28	CD4+ T cells (50%)	CD80 (B7.1), CD86 (B7.2)	Akt, PI3K, Bcl-xL	Enhancer
ICOS (CD278)	CD8+ T cells (80%)	ICOSL (CD275)	PI3K, Akt, MAPK	Enhancer
CTLA-4 (CD152)	γδ T cells, activated B cells, DCs, macrophages	CD80 (B7.1), CD86 (B7.2)	SHP-2, PI3, PP2A, Akt, JNK, ERK	Silencer
PD-1 (CD279)	NK cells, naive and memory T cells, Tregs	PD-L1, CD274, B7-H1	SHP-2. PI3, Akt, ZAP-70, PKCθ, ERK, STAT5	Silencer
BTLA	Tregs, active CD4+ T cells, B cells, NK cells, DCs, granulocytes, myeloid-derived suppressor cells (MDSCs)	HVEM, CD160, LIGHT	BMP, MAPK, DDX19B, IL17RA	Silencer

**Table 2 ijms-25-01274-t002:** The list of checkpoint inhibitors.

Drug	Brand Name	Target Molecule	Medical Uses	FDA Approval
Toripalimab	Loqtorzi	PD-1	Nasopharyngeal carcinoma	2023
Cemiplimab	Libtayo	PD-1	Metastatic cutaneous squamous cell carcinoma	2018
Durvalumab	Imfinzi	PD-L1	Urothelial carcinoma	2017
Avelumab	Bavencio	PD-L1	Merkel cell carcinoma, urothelial carcinoma	2017
Atezolizumab	Tecentriq	PD-L1	Urothelial carcinoma, non-small cell lung cancer, small cell lung cancer, hepatocellular carcinoma, urothelial carcinoma	2016
Pembrolizumab	Keytruda	PD-1	Melanoma and Hodgkin’s lymphoma	2014
Nivolumab	Opdivo	PD-1	Metastatic melanoma, non-small cell lung cancer, renal cell carcinoma, hepatocellular carcinoma, esophageal cancer, gastric cancer, Hodgkin’s lymphoma	2014
Ipilimumab	Yervoy	CTLA-4	Melanoma	2011
Abatacept	Orencia	CD80 and CD86	Rheumatoid arthritis, juvenile idiopathic arthritis, psoriatic arthritis	2005

## Data Availability

Data are contained within the article.

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
