# Peer review of "The Role of the CD28 Family Receptors in T-Cell Immunomodulation"

_ijms, 2024, doi:10.3390/ijms25021274_

Round 1

Reviewer 1 Report

Comments and Suggestions for Authors

This paper reviewed the molecular basis of the functioning of CD28 family receptors (including CD28, ICOS, CTLA-4, PD-1, and BTLA molecules) and their impact on T-cell activity. The topic fits the scope of this journal and may benefit the understanding of the adaptive response, as well as the development of immune-related diseases. In general, this manuscript is well-organized and the references can support the conclusions. The references are also updated. Key issues are required to be addressed before its publication on Int. J. Mol. Sci.

1. The drug discovery efforts targeting the CD28 family receptors are suggested to be included with citing relevant reviews or key research papers to further broaden the readership of this manuscript.

2. The drugs targeting the members of CD28 family receptors are suggested to be briefly summarized in a table with the related activities and the current development statues.

3. The future perspectives of these CD28 family receptors are suggested to be further discussed in this manuscript.

Comments on the Quality of English Language

The English language is fine.

Author Response

"This paper reviewed the molecular basis of the functioning of CD28 family receptors (including CD28, ICOS, CTLA-4, PD-1, and BTLA molecules) and their impact on T-cell activity. The topic fits the scope of this journal and may benefit the understanding of the adaptive response, as well as the development of immune-related diseases. In general, this manuscript is well-organized and the references can support the conclusions. The references are also updated. Key issues are required to be addressed before its publication on Int. J. Mol. Sci."

Thank you for all your comments and suggestions. Corrections in the text are marked in green.

"The drug discovery efforts targeting the CD28 family receptors are suggested to be included with citing relevant reviews or key research papers to further broaden the readership of this manuscript."

Thank you for the suggestion. We extended a paragraph about the importance of CD28 family receptors (page 9, lines 384-409): "Among the antibodies targeting PD-L1, we have avelumab (Bavencio), and durvalumab (Imfinzi) registered for patients with urothelial carcinoma [92]. The list of ICIs and the diseases for which they are approved is presented in Table 2.

For autoimmune diseases, there is only one checkpoint stimulator, abatacept, a CTLA4-Ig fusion protein, one of the therapeutic options for rheumatoid arthritis. Abatacept attenuates T-cell activation because it regulates T-cell activation by inhibiting the CD80/86-CD28 co-stimulatory pathway, which is required for normal T-cell activation (Table 2) [94].

The importance of CD28 co-stimulatory signals in T-cell function makes this molecule an exciting target for drug design to modulate the function of both effector T cells and Treg cells. Unfortunately, the first attempts to create a soluble CD28 (sCD28) to block co-stimulatory signals failed due to the too-low affinity of sCD28 for ligands [95]. Considering the known mechanisms, attempts are still being made to test available drugs for the treatment of autoimmune diseases [96, 97, 98], including type 1 diabetes [99], lupus nephritis [100], asthma [101], or Crohn's disease [102].

Understanding molecular mechanisms responsible for regulating immune checkpoints may constitute the basis for new immunotherapies. Using small molecules, such as peptides, as adjuvants is another exciting option, as these compounds have high selectivity and potency, better tolerability, and predictable metabolism. An example is glycoprotein D, which can bind with HVEM and block its interactions with BTLA [103]. Another advantage of using ICIs is the possibility of using them in patients who do not qualify for clinical trials for various reasons, for example, cancer patients with pre-existing autoimmune diseases, who are usually excluded from immunotherapy clinical trials due to the high risk of serious adverse events. There is growing evidence that immune checkpoint inhibitors may be safe and effective in this patient population [104]."

"The drugs targeting the members of CD28 family receptors are suggested to be briefly summarized in a table with the related activities and the current development statues."

As suggested, we added Table 2.

Drug

Brand name

Target molecule

Medical uses

FDA approval

Toripalimab

Loqtorzi

PD-1

nasopharyngeal carcinoma

2023

Cemiplimab

Libtayo

PD-1

metastatic cutaneous squamous cell carcinoma

2018

Durvalumab

Imfinzi

PD-L1

urothelial carcinoma

2017

Avelumab

Bavencio

PD-L1

Merkel cell carcinoma, urothelial carcinoma

2017

Atezolizumab

Tecentriq

PD-L1

urothelial carcinoma, non-small cell lung cancer, small cell lung cancer, hepatocellular carcinoma, urothelial carcinoma

2016

Pembrolizumab

Keytruda

PD-1

melanoma, Hodgkin's lymphoma

2014

Nivolumab

Opdivo

PD-1

metastatic melanoma, non-small cell lung cancer, renal cell carcinoma, heaptocellular carcinoma, esophageal cancer, gastric cancer, Hodgkin's lymphoma

2014

Ipilimumab

Yervoy

CTLA-4

melanoma

2011

Abatacept

Orencia

CD80 and CD86

rheumatoid arthritis, juvenile idiopathic arthritis, psoriatic arthritis

2005

"The future perspectives of these CD28 family receptors are suggested to be further discussed in this manuscript."

According to the Reviewer's suggestions, we added a paragraph in the conclusions: “The central importance of CD28 family receptors in T-cell function makes it a tempting target for drugs that modulate the function of both effector T cells and Tregs. These signals are crucial in many T-cell processes, including cytoskeletal remodeling, cytokine production, survival, and differentiation. CD28 and ICOS ligation leads to unique epigenetic, transcriptional, and post-translational changes in T cells that cannot be reproduced by TCR ligation alone, making them crucial for T-cell survival, activation, and maintenance of immune homeostasis. However, the second aspect of controlling T-cell activation is inhibiting the processes that may contribute to immune-mediated tissue damage.

By modulating the length and magnitude of immune responses, immune checkpoint molecules keep immune homeostasis. Disrupting their function impairs the control of T-cell responses, which may promote the development of autoimmune diseases or cancer. Genetic polymorphisms in human immune checkpoint molecules account for additional variability in this system. Therefore, targeting the receptors or ligands of immune checkpoints using agonistic or antagonistic antibodies can alter immune tolerance and explain the likelihood of adverse reactions and their mechanisms. Expanding knowledge of the function of immune checkpoint molecules and the impact of their polymorphisms can bring information on how to precisely and deliberately modulate T-cell activation pathways to either suppress pathological responses promoting auto-immunity or activate the anti-tumor response.” (page 10, lines 414-432).

Reviewer 2 Report

Comments and Suggestions for Authors

 The paper is interesting and well-written but can not be published in this present form.

MAJOR QUESTION

1.       The main problem with this review is that the references are obsolete, the obslescence index is 18 years (2023- Median Publication date of bibliografic references).  This is to obsolete for a review paper in basic sciences. The authors should do a review in deep and use updated bibliografly

MINOR ISSUES

2.     The paper should contemplate how molecular interactions in T-cell activation translate to practical immune responses

3.       There is a a great variability in the discussin of diferents molecule eg: CD28 and CTLA-4 are studied in deep while  PD-1 are less detailed.

4.     The authos said that BTLA is not found in regulatory T cells (Tregs), which contradicts some other studies suggesting its expression in Tregs. Authos should discuss the literature and the conflictive findigns.

Papers taht said that there is the presence of BTLA in Tregs:

 https://doi.org/10.3389/fimmu.2021.767099

https://doi.org/10.1016/j.celrep.2022.11055

Examples of papers that Supporting the absence of BTLA in Tregs:

https://doi.org/10.4049/jimmunol.1501973

https://doi.org/10.1016/j.immuni.2016.10.030

5.       Provide mor information about how the interaction  between BTLA and herpes virus entry mediator impacts immune regulation

Author Response

"The main problem with this review is that the references are obsolete, the obslescence index is 18 years (2023- Median Publication date of bibliografic references). This is to obsolete for a review paper in basic sciences. The authors should do a review in deep and use updated bibliography."

We updated references according to Reviewer's suggestions. Corrections in the text are marked in red.

"The paper should contemplate how molecular interactions in T-cell activation translate to practical immune responses."

According to the Reviewer's suggestions, we added a paragraph about the potential practical benefits of controlling the process of T-cell activation: "The central importance of CD28 family receptors in T-cell function makes it a tempting target for drugs that modulate the function of both effector T cells and Tregs. These signals are crucial in many T-cell processes, including cytoskeletal remodeling, cytokine production, survival, and differentiation. CD28 and ICOS ligation leads to unique epigenetic, transcriptional, and post-translational changes in T cells that cannot be reproduced by TCR ligation alone, making them crucial for T-cell survival, activation, and maintenance of immune homeostasis. However, the second aspect of controlling T-cell activation is inhibiting the processes that may contribute to immune-mediated tissue damage.

By modulating the length and magnitude of immune responses, immune checkpoint molecules keep immune homeostasis. Disrupting their function impairs the control of T-cell responses, which may promote the development of autoimmune diseases or cancer. Genetic polymorphisms in human immune checkpoint molecules account for additional variability in this system. Therefore, targeting the receptors or ligands of immune checkpoints using agonistic or antagonistic antibodies can alter immune tolerance and explain the likelihood of adverse reactions and their mechanisms. Expanding knowledge of the function of immune checkpoint molecules and the impact of their polymorphisms can bring information on how to precisely and deliberately modulate T-cell activation pathways to either suppress pathological responses promoting auto-immunity or activate the anti-tumor response.” (page 10, lines 414-432).

"There is a great variability in the discussin of diferents molecule eg: CD28 and CTLA-4 are studied in deep while  PD-1 are less detailed."

According to Reviewer's suggestions, we expanded this paragraph: "Interactions between PD-1 and its ligands transmit an inhibitory signal to T cells, which has a measurable effect of hampering the proliferation and production of cytokines by these cells. [69]. However, studies show that stimulating PD-L1 can induce the conversion of naive CD4+ T cells to induced T regulatory cells (iTregs), which are critical mediators of peripheral tolerance that actively suppress the formation of effector T (Teff) cells [70]. In the absence of PD-1 signaling, PD-L1 and PD-L2 may provide a positive signal to T cells and stimulate their proliferation and cytokine production [71]. Teff cells, when activated, express both PD-1 and PD-L1. Also, higher PD-L1 expression is correlated with a higher Teff proliferation capacity [72]." (page 7, lines 311-396).

"The authors said that BTLA is not found in regulatory T cells (Tregs), which contradicts some other studies suggesting its expression in Tregs. Author should discuss the literature and the conflictive findings.

 Papers that said that there is the presence of BTLA in Tregs:

https://doi.org/10.3389/fimmu.2021.767099

https://doi.org/10.1016/j.celrep.2022.11055

Examples of papers that Supporting the absence of BTLA in Tregs:

https://doi.org/10.4049/jimmunol.1501973

https://doi.org/10.1016/j.immuni.2016.10.030"

As suggested, we added a sentence about contradicting findings: "However, opinions are divided on the presence of the receptor on Tregs. While some argue that BTLA is found in Tregs [73, 74], others deny it. [75, 76, 77]." (page 7, lines 325-374).

"Provide more information about how the interaction  between BTLA and herpes virus entry mediator impacts immune regulation."

As suggested, we added a paragraph about it: "Its binding with BTLA reduces cell activation, cytokine production, and proliferation [81]. Therefore, BTLA deficiencies in various experimental animal models promote the development of autoimmune diseases or worsen their course [82, 83]. For example, BTLA-deficient mice presented more pronounced experimental autoimmune encephalomyelitis. It should also be emphasized that HVEM can interact with BTLA in a cis or trans manner. HVEM and BTLA can be co-expressed on the same cells, forming a cis complex, or present on different cells, forming a trans interaction [84]. Both types of interaction inhibit T-cell activity but in different mechanisms; the cis complex prevents interaction with other co-signaling molecules, while trans interaction inhibits nuclear factor kappa-light-chain-enhancer of activated B cells (NF-κB)-dependent cell activation." (page 8, lines 336-346).

Round 2

Reviewer 2 Report

Comments and Suggestions for Authors

The authors have incorporated all the sugesstion of the previous report. They have made a great work the paper can be published.